# Population-based incidence and antimicrobial susceptibility patterns of shigellosis among children and adults from rural and urban Kenya, 2010–2019

Richard Omore[1]*, Billy Ogwel[1], John B. Ochieng[1], Jane Juma[1], Victor Omballa[1], Alice Ouma[1], George Aol[1], Allan Audi[1], George O. Agogo[2], George Odongo[3], Clayton Onyango[3], Newton Wamola[1], Terry Komo[1], Daisy Chepkemoi[1], Elizabeth Hunsperger[3], Daniel R. Feikin[3,4], Joel M. Montgomery[3,5], Marc-Alain Widdowson[3], Graeme Prentice-Mott[6], Eric D. Mintz[6], Robert F. Breiman[3,7], Patrick K. Munywoki[3], Godfrey M. Bigogo[1], Jennifer R. Verani[3]

1 Kenya Medical Research Institute-Center for Global Health Research (KEMRI-CGHR), Kisumu, Kenya, 2 Division of Global Health Protection (DGHP), Kenya Office of the US Centers for Disease Control and Prevention, Nairobi, Kenya, 3 Division of Global Health Protection, Centre for Global Health, US Centers for Disease Control and Prevention, Atlanta, Georgia, United States of America, 4 Independent Consultant, Coppet, Switzerland, 5 National Center for Emerging and Zoonotic Infectious Diseases (NCEZID), Centers for Disease Control and Prevention, Atlanta, Georgia, United States of America, 6 Division of Foodborne, Waterborne, and Environmental Diseases, Centers for Disease Control and Prevention, Atlanta, Georgia, United States of America, 7 Department of Global Health, Rollins School of Public Health, Emory University, Atlanta, Georgia, United States of America

* omorerichard@gmail.com

## Abstract

### Backgroun

*Shigella* is an important cause of diarrheal morbidity and mortality globally. Data on disease burden across age groups, in different epidemiologic settings, and over time are needed to guide preventive strategies. We examined shigellosis in two sites in Kenya over a 10-year period.

### Methods

We used data from the Population-Based Infectious Disease Surveillance (PBIDS) platform in a rural (Asembo, population ~35,000) and urban (Kibera, population ~23,000) setting. PBIDS participants presenting to surveillance clinics with diarrhea (≥3 loose stools in 24-hour period) had stool collected and cultured; *Shigella* isolates underwent antimicrobial susceptibility testing. We estimated incidence by dividing *Shigella* cases by person-years-observation, adjusting for the proportion of diarrhea cases with stool collected and for care-seeking outside surveillance clinics.

**Data availability statement:** The data used in this study are owned by the Kenya Medical Research Institute (KEMRI). Access to the data may be granted upon reasonable request to the program (info.pbids@kemri.go.ke) or Head of the KEMRI Scientific and Ethics Review Unit (Email: seru@kemri.go.ke or serukemri@gmail.com), subject to KEMRI's institutional data access and sharing policies.

**Funding:** This study was funded by the US Centers for Disease Control and Prevention (CDC) in Atlanta USA through a cooperative agreement with KEMRI. The funders had no role in study design, data collection and analysis, decision to publish, or preparation of the manuscript.

**Competing interests:** The authors declare no competing interest.

## Results

From January 1, 2010 to December 31, 2019, we isolated *Shigella* from 23% and 15% of 2,017 and 4,074 stool specimens collected in Asembo and Kibera, respectively; *S. flexneri* accounted for 61% and 67%, respectively. In Asembo, the adjusted shigellosis incidence was 684/100,000; it was highest in ages 12–23 months (1,873/100,000) and ≥50 years (1,502/100,000). In Kibera, the adjusted incidence was 647/100,000, highest in ages 12–23 (2,828/100,000) and 24–59 months (936/100,000). Incidence declined significantly in Asembo (p = 0.009), but not in Kibera (p = 0.53). Overall, ≥97% of isolates were susceptible to ciprofloxacin and ceftriaxone.

## Conclusion

The shigellosis burden was greatest among young toddlers in both urban and rural areas and was high among older adults in the rural setting. Although resistance to first-line antibiotics was infrequent, continued susceptibility monitoring is warranted.

## Background

*Shigella* is a common cause of diarrheal morbidity and mortality in lower-middle-income countries [1,2]. *Shigella* is highly infectious and spreads through the fecal-oral route with high burden in areas with poor sanitation [3,4]. While all four species of *Shigella* (Serogroup A: S. *dysenteriae*, Serogroup B: *S. flexneri*, Serogroup C: *S. boydii*, and Serogroup D: *S. sonnei*) can cause diarrhea and dysentery, species vary in disease severity and geographical distribution [5].

Efforts to prevent and control shigellosis have focused primarily on improvements in safe water, sanitation, and hygiene (WASH), promotion of exclusive breastfeeding for the first six months of life, and prompt treatment with an effective antimicrobial agent to decrease duration of illness and spread of infection [6]. However, the disease burden remains high in settings with limited access to sanitation and healthcare [1,2]. Furthermore, the emergence of resistance among *Shigella* isolates to commonly used antibiotic agents limits therapeutic options for bacteria in some settings [7]. No vaccines for shigellosis are currently licensed, but several candidates are in development and may offer a complementary strategy to reduce disease burden [1].

Information on *Shigella* disease burden across age groups, in different epidemiologic settings, and over time is needed to guide vaccine development and other preventive strategies. We used population-based surveillance data to characterize the epidemiology and antibiotic susceptibility patterns of shigellosis in a rural and an urban setting in Kenya over a 10-year period.

## Materials and methods

### Study setting and study population

The Kenya Medical Research Institute (KEMRI) in collaboration with the US Centers for Disease Control and Prevention (CDC) has jointly operated the Population-Based Infectious Disease Surveillance (PBIDS) platform since 2006 in Asembo and Kibera; the study methods have been detailed elsewhere [8,9]. Both settings have high burdens of infectious disease, characterized by inadequate WASH, and human colonization with multi-drug resistant bacteria is common [8,10,11]. Kibera is a large, densely populated urban informal settlement in Nairobi (population ~23,000) while Asembo is a sparsely populated rural area (population ~35,000) that is malaria-endemic [8] necessitating careful interpretation of fever within our syndromic diarrhea surveillance. The estimated HIV prevalence in Asembo and Kibera among adults was 16% and 6% in 2012 [12] and 15% and 4% in 2018–2019, respectively [13]. Persons residing in the surveillance areas for ≥ four consecutive months were eligible for participation in PBIDS surveillance including children born to active PBIDS participants [8,9]. S1 Fig shows the population distribution of the PBIDS residents at the community and sentinel clinics in both sites.

### Data collection methods

Enrolled PBIDS households were visited regularly for collection of demographic data and recent illness including diarrhea (≥3 looser than normal stools in a 24-hour period) [8]. If diarrhea was reported, additional data were collected on whether the stool was bloody, whether healthcare was sought and if so, at which health facility. The frequency of household visits at the start of this study period (January 1, 2010) was weekly. In May 2011, the frequency changed to biweekly, and in April 2015, to biannual. The household data collection tools remained unchanged, and the biannual rounds were conducted continuously over the course of the year.

PBIDS participants received free medical care for acute infectious illnesses at a centrally located health facility in each site. In Asembo, participants resided within ~5 kilometers of the surveillance health facility— St. Elizabeth Lwak Mission Hospital, which has a large outpatient clinic and small inpatient ward (~60 beds). In Kibera, PBIDS participants resided within ~1 kilometer of Tabitha Medical Clinic, which provided outpatient services only. At each of the surveillance facilities (Fig 1), trained clinical staff assessed patients and entered patient data into an electronic patient care system. Patients presenting with diarrhea (≥3 loose stools or clinician diagnosis of diarrhea) were eligible for stool sample collection. Characteristics of stool were captured based on patient or caretaker report.

### Laboratory methods

Whole stool specimens were collected in stool cups and immediately placed in Cary-Blair transport media, maintained at 4° C, and transported within 24 hours to KEMRI microbiology laboratories located near each site [8]. Stool specimens were cultured by standard techniques for bacterial pathogens; *Shigella* was identified biochemically [11]. Susceptibility of *Shigella* isolates to a panel of antimicrobial agents was determined by Kirby Bauer disk diffusion method and interpreted according to 29th edition Clinical and Laboratory Standards Institute (CLSI) guidelines [14]. Isolates that were resistant or had intermediate resistance to antibiotics tested were classified as non-susceptible. Laboratory results were communicated back to the surveillance facilities to guide clinicians in patient management.

### Statistical analysis

We described the demographic and clinical characteristics of shigellosis cases using frequencies and percents. Crude shigellosis incidence was calculated as the number of culture-confirmed *Shigella* cases divided by person-years of observation (PYO) based on cumulative residency within the PBIDS area from January 1, 2010, to December 31, 2019, accounting for migrations. We employed a two-step adjustment in calculating adjusted incidence, to account for both under-sampling and out-of-network care seeking by weighting the number of observed culture-confirmed cases by two

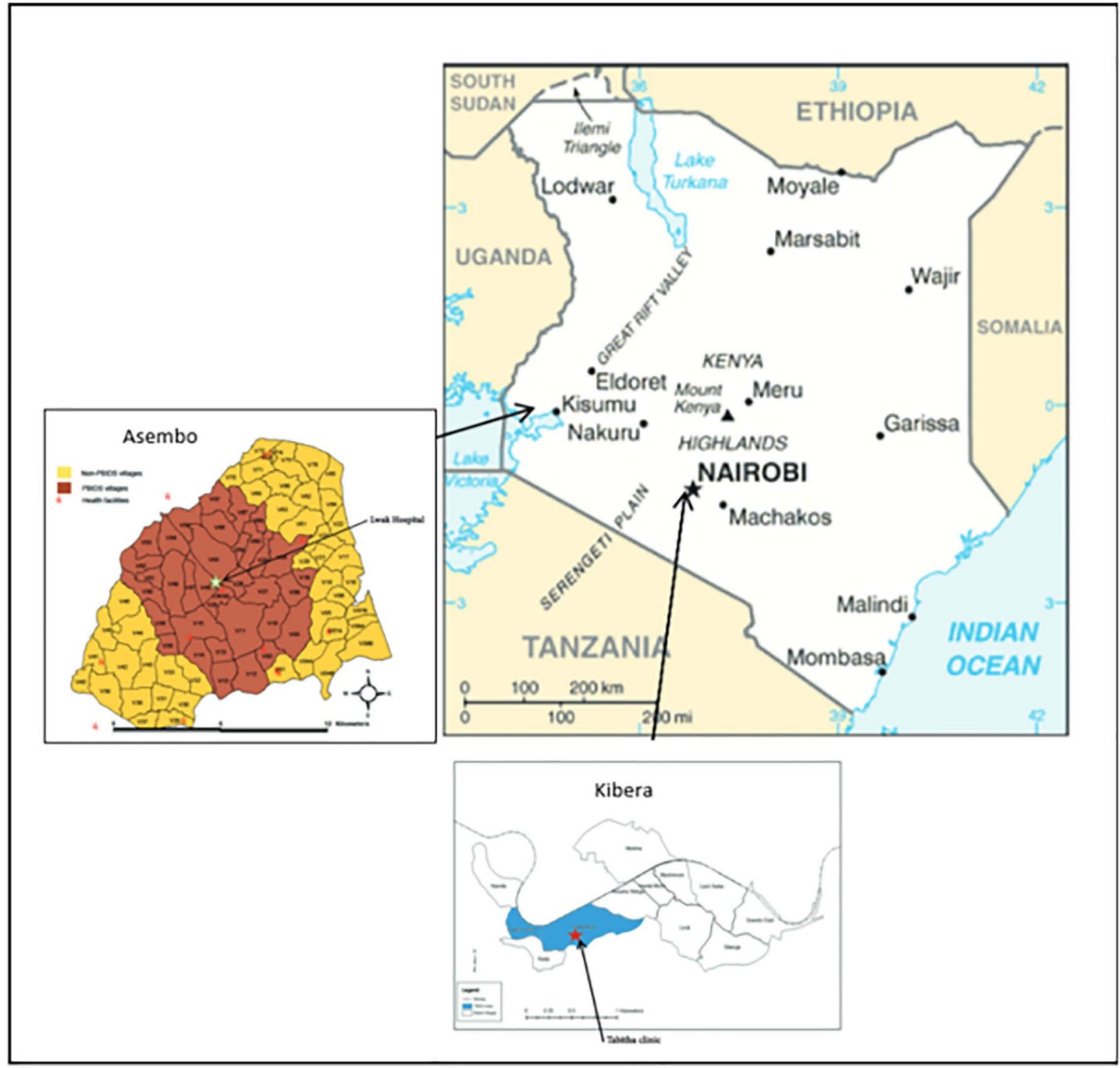

**Fig 1. A series of maps showing Kenya's location in East Africa, and the locations of KEMRI and CDC jointly operated PBIDS study areas served by St. Elizabeth Lwak Mission Hospital in Siaya County and Tabitha Medical clinic in Kibera, Nairobi, Kenya.**

proportions: [1] the proportion of PBIDS participants eligible for stool collection (presenting to the clinic with the diarrhea case definition) who had stool collected, and [2] the proportion of diarrheal episodes reported, at household visits, to have sought medical care at a health facility and for which this care was sought at the study facility.

Chi-square or Fisher's exact tests were used to compare characteristics of participants who had stool collected versus those who did not to understand potential limitations of the adjustment for sample collection. Adjustment factors were stratified by site, age group, study year, and whether diarrhea was bloody or not.

Adjusted case counts were divided by the respective PYOs to estimate adjusted incidence. The uncertainty around the adjusted incidence was calculated using 95% Confidence Intervals (CI) estimated via Monte Carlo simulations (0.025 and 0.975 quantiles of 10,000 simulations), sampling from Poisson distribution for crude incidence, and binomial distribution for adjustment factors as detailed elsewhere by Greenland [15], and as applied previously to data from this surveillance platform [16]. Trends in adjusted shigellosis incidence were assessed using the Mann-Kendall test. For all statistical tests, a two-sided p-value of <0.05 was considered statistically significant. Analyses were performed using SAS statistical software version 9.4 (SAS Institute Inc., Cary, NC).

## Ethical procedures

The protocols for this study were reviewed and approved by KEMRI Scientific and Ethics Review Unit (KEMRI SSC Protocol # 1899 and 2761) and CDC (protocol # 4566 and #6775) (See 45 C.F.R. part 46; 21 C.F.R. part 56). Written informed consent was provided by heads of households for household-level participation in PBIDS; individual household members could decline participation. Individual written informed consent was obtained from all patients (or their parents/guardians if aged <18 years) before specimen collection at the surveillance facilities.

## Results

### Clinic visits and sample collection

From January 1, 2010, through December 31, 2019, there were 171,466 and 176,481 clinic visits in Asembo and Kibera, respectively. of whom 6,495 (4%) and 8,498 (5%) met the case definition for diarrhea, respectively (Fig 2). Overall, stool specimen was collected from 2,017 (31%) of diarrhea cases in Asembo and 4,074 (48%) in Kibera. In both sites, diarrhea cases who provided a stool specimen were older and more frequently presented with bloody stool than those who did not provide stool; in Asembo, additional factors associated with stool specimen collection were presenting with sunken eyes, and hospitalization with male sex less likely to produce stool specimen (S1Table).

### Characteristics of shigellosis cases

We isolated *Shigella* from 23% of stool specimens collected in Asembo, and from 15% in Kibera (Fig 2). Among shigellosis cases in Asembo and Kibera, the median age was 33 and 20 years, respectively; 34% and 44%, respectively, were male (Table 1). Among *Shigella* cases with known HIV status in Asembo (64%) and Kibera (9%), the HIV infection rate was 25% (74/296) and 9% (5/53), respectively. Bloody stool was reported in 47% of cases in Asembo and 36% in Kibera. In both sites, the most common *Shigella* species was *S. flexneri*, accounting for 61% of isolates in Asembo and 67% in Kibera (Table 1). The relative frequency of species varied annually over the 10-year study period in both Asembo and Kibera but with no clear trend over time (S2 Table). Specifically, in both Asembo and Kibera, *S. flexneri* consistently dominated over the years (42%−90%), with *S. sonnei* showing a recent upward trend. After a 2013 peak, *S. dysenteriae* declined in Asembo, while remaining low in Kibera. and *S. boydii* was more variable in Kibera compared to a low and sporadic presence in Asembo.

### Incidence rates of shigellosis

The overall crude incidences for shigellosis in Asembo and Kibera surveillance areas were 137 (95%CI 66–499) and 239 (95%CI 121–866) per 100,000 PYO, respectively; adjusted incidences were 684 (95%CI 151–2,332) and 647 (95% CI 288-3,887) per 100,000 PYO, respectively (S3 and S4 Tables). In Asembo, the adjusted incidence was highest in the 12–23 months age group (1,873; 95%CI 1126–2937), followed by ≥50 years (1,502; 95%CI 1,202-1,979), and 35–49

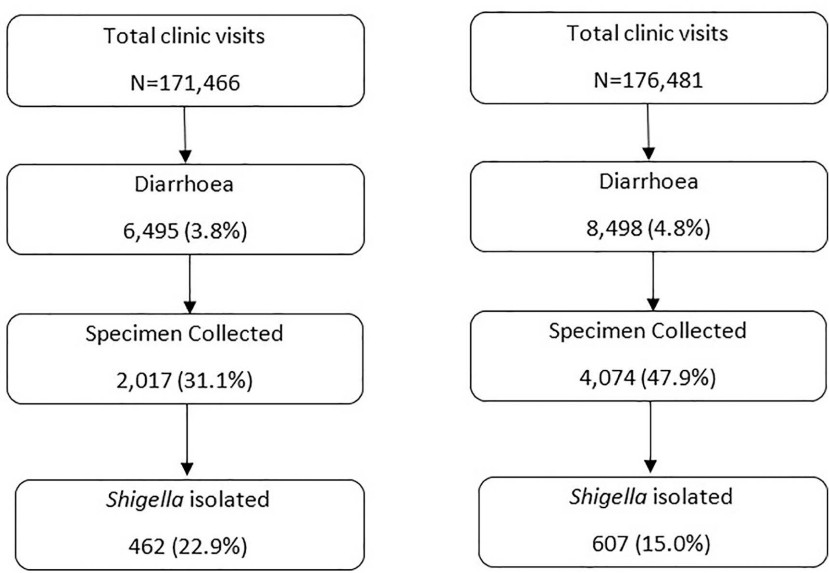

**Lwak Mission Hospital, Asembo**

Total clinic visits
N=171,466

↓

Diarrhoea
6,495 (3.8%)

↓

Specimen Collected
2,017 (31.1%)

↓

*Shigella* isolated
462 (22.9%)

**Tabitha Medical Clinic, Kibera**

Total clinic visits
N=176,481

↓

Diarrhoea
8,498 (4.8%)

↓

Specimen Collected
4,074 (47.9%)

↓

*Shigella* isolated
607 (15.0%)

**Fig 2. Flow diagram of diarrhoea cases and *Shigella* testing among Patients seeking care from PBIDS 2010-2018.**

years (1,068; 95%CI 818-1,471) (Fig 3 and S3 Table). In Kibera, the adjusted incidence was also highest in the 12–23 months age group (2,828; 95% CI 1,527-6,074) followed by 24–59 months (936; 95%CI 692-1,271), and <12 months (698; 95%CI 127–12,250). By year, the highest adjusted incidence in Asembo was in 2011 (1,227; 95%CI 826-1,991) and the lowest in 2016 (275; 95%CI 158–432) (Fig 4 and S4 Table). In Kibera, the adjusted incidence was highest in 2012 (1,330; 95%CI 923-2,225) and lowest in 2017 (213; 95%CI 122–324). We did not identify any clear outbreak or surveillance change explaining these peaks. In Asembo the adjusted incidence declined significantly during the study period (p = 0.0095), but there was no trend over time in Kibera (p = 0.5312).

## Antimicrobial prescription and susceptibility patterns

Among shigellosis cases in Asembo, 74% (343/462) were treated with antibiotics on initial presentation to the clinic, including 75% (163/218) of those with and 74% (180/244) without bloody diarrhea (Table 1). Among those prescribed an antibiotic, the most common agents were nalidixic acid (62%), metronidazole—likely prescribed empirically for undifferentiated diarrhea—(46%), and ciprofloxacin (22%). In Kibera, 46% (279/607) of cases were treated with antibiotics, including 43% (92/216) and 48% (187/391) of those with and without bloody diarrhea, respectively. The most frequently used antibiotics in Kibera were ciprofloxacin (67%), metronidazole (48%), and erythromycin (15%).

Only 2% of patients received antibiotics recommended for *Shigella* to which their isolates were non-susceptible, suggesting generally appropriate empiric treatment. In both sites, *Shigella* isolates were most frequently non-susceptible to sulfisoxazole, trimethoprim, streptomycin, tetracycline, and ampicillin (Fig 5). In Kibera, 3% of isolates were non-susceptible to ciprofloxacin and 1% ceftriaxone; in Asembo, 1% were non-susceptible to ciprofloxacin and 2% to ceftriaxone. The frequency of ciprofloxacin and ceftriaxone non-susceptibility were highest in 2016 (2/24 [8%]) and 2010 (2/35 [6%]), respectively, in Asembo and 2013 (4/47 [9%]) and 2015 (2/71 [3%]), respectively, in Kibera. Although rare, one isolate in Kibera in 2013 was resistant to both ciprofloxacin and ceftriaxone—the two primary recommended treatments for shigellosis. No isolates from Asembo were non-susceptible to both agents during the study period. In Asembo,

**Table 1. Characteristics of medically attended *Shigella* cases in Lwak Mission Hospital, Asembo and Tabitha Medical Clinic, Kibera, Kenya, 2010-2019.**

|  | Asembo (n = 462) | Kibera (n = 607) | Overall (n = 1,069) |
|---|---|---|---|
| Characteristics | n (%) | n (%) | n (%) |
| Age |  |  |  |
| <12m | 8 (2) | 7 (1) | 15 (2) |
| 12-23m | 24 (5) | 31 (5) | 55 (5) |
| 24-59m | 30 (7) | 67 (11) | 97 (9) |
| 5-9yrs | 15 (3) | 70 (12) | 85 (8) |
| 10-17yrs | 49 (11) | 110 (18) | 159 (15) |
| 18-34yrs | 117 (25) | 207 (34) | 324 (30) |
| 35-49yrs | 81 (17) | 89 (15) | 170 (16) |
| 50 + yrs | 138 (30) | 26 (4) | 164 (15) |
| Male | 157 (34) | 269 (44) | 426 (40) |
| HIV Status |  |  |  |
| Known | 296 (64) | 53 (9) | 349 (33) |
| Unknown | 166 (36) | 554 (91) | 720 (67) |
| HIV Status Known |  |  |  |
| HIV Positive | 74 (25) | 5 (9) | 79 (23) |
| HIV Negative | 222 (75) | 48 (91) | 270 (77) |
| Species |  |  |  |
| *S. flexneri* | 280 (61) | 404 (67) | 684 (64) |
| *S. sonnei* | 72 (15) | 68 (11) | 140 (13) |
| *S. dysenteriae* | 47 (10) | 41 (7) | 88 (8) |
| *S. boydii* | 40 (9) | 46 (7) | 86 (8) |
| Non-typable | 23 (5) | 48 (8) | 71 (7) |
| Clinical features |  |  |  |
| Dehydration | 4/62 (7) | 1/105 (1) | 5/167 (3) |
| Sunken eyes | 11 (2) | 2 (0) | 13 (1) |
| Fever | 259 (56) | 112 (19) | 371 (35) |
| Vomiting | 113 (25) | 52 (9) | 165 (15) |
| Lethargic | 7 (2) | 3 (1) | 10 (1) |
| Convulsion | 1 (0) | 0 (0) | 1 (0) |
| Watery/Mucus in stool | 373 (81) | 299 (49) | 672 (63) |
| Bloody stool | 218 (47) | 216 (36) | 434 (41) |
| Treatment |  |  |  |
| Sought Care before coming to clinic | 196 (42) | 156 (26) | 352 (33) |
| Reported taking an antibiotic | 49 (11) | 38 (6) | 87 (8) |
| Hospitalized | 82 (18) | 0 (0) | 82 (8) |
| Treat with antibiotic at health facility | 343 (74) | 279 (46) | 622 (58) |
| Nalidixic acid | 180/343 (53) | 6/279 (2) | 186/622 (30) |
| Metronidazole | 175/343 (51) | 152/279 (55) | 327/622 (53) |
| Ciprofloxacinᵝ | 71/343 (21) | 128/279 (46) | 199/622 (32) |
| Erythromycin | 15/343 (4) | 67/279 (24) | 82/622 (13) |
| Gentamycin | 11/343 (3) | 1/279 (0) | 12/622 (2) |
| Ceftriaxoneᵝ | 9/343 (3) | – | 9/622 (1) |
| Ampicillin | – | 8/279 (3) | 8/622 (1) |
| Treated with Antibiotic to which their isolates were resistant | 7/343 (2) | 6/279 (2) | 13/622 (2) |

*(Continued)*

**Table 1.** (Continued)

β-Antibiotics recommended for treatment of shigellosis.

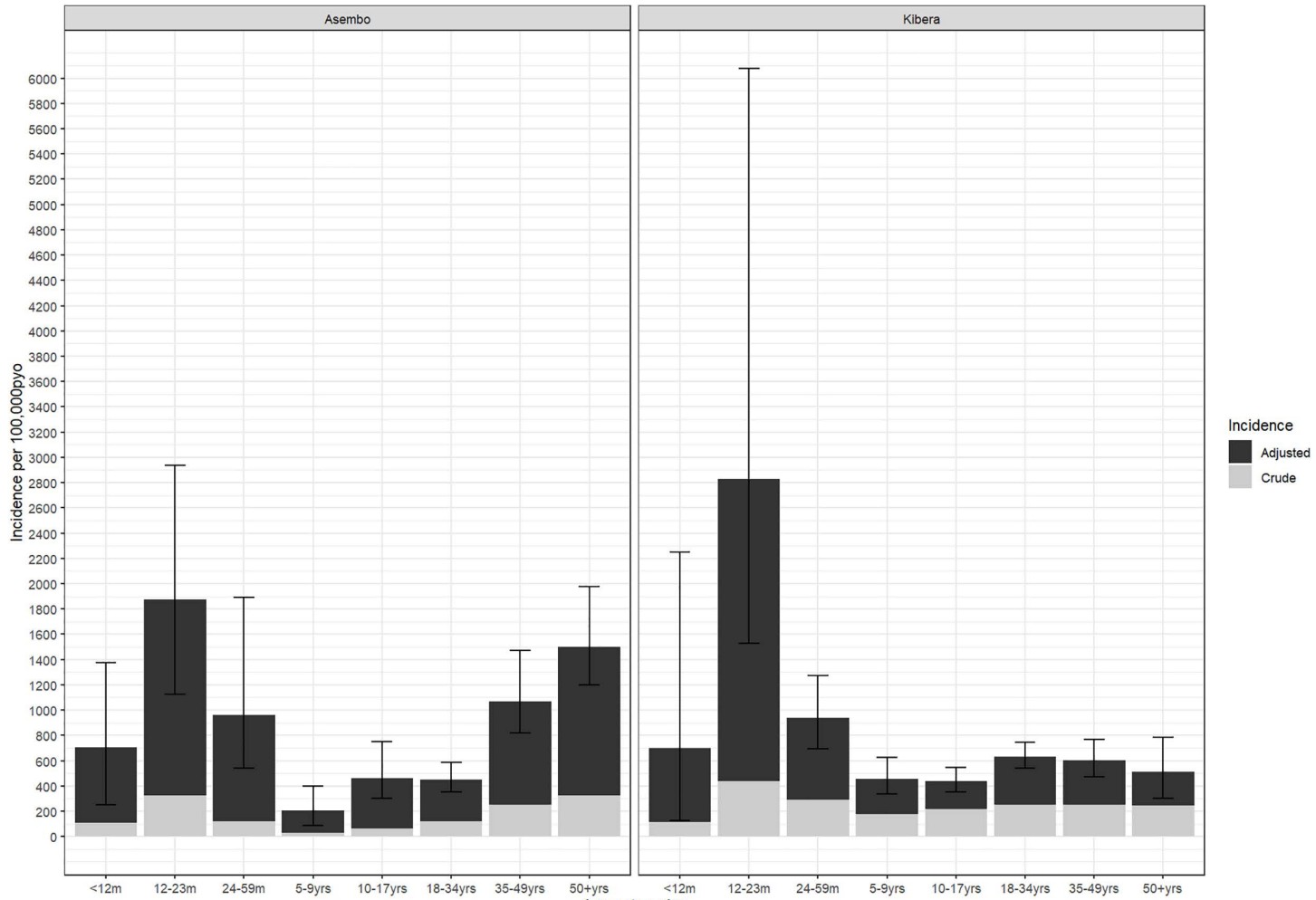

**Fig 3. Crude and Adjusted Age-Stratified Shigella Incidence in PBIDS: 2010-2018.** *Incidence rates were adjusted first for the proportion who were sampled among diarrhea cases, and second for proportion of cases with diarrhea at home who sought care in any clinic other than the surveillance clinic.

non-susceptibility did not significantly change over the study period (Fig 6). In contrast to other antibiotics, nalidixic acid, non-susceptibility in Kibera increased significantly from 8% to 25% (p = 0.0007).

## Discussion

Our study findings demonstrated: 1) a high incidence of shigellosis in both rural and urban areas of Kenya, with children aged 12–23 months bearing the greatest burden in both areas, and older adults also heavily affected in the rural area; 2) declining incidence over time in the rural setting, but not in the urban setting; 3) *Shigella* isolates were largely susceptible to first line antibiotics for treating shigellosis; and 4) *S. flexneri* accounted for approximately two-thirds of all *Shigella* isolates in both sites.

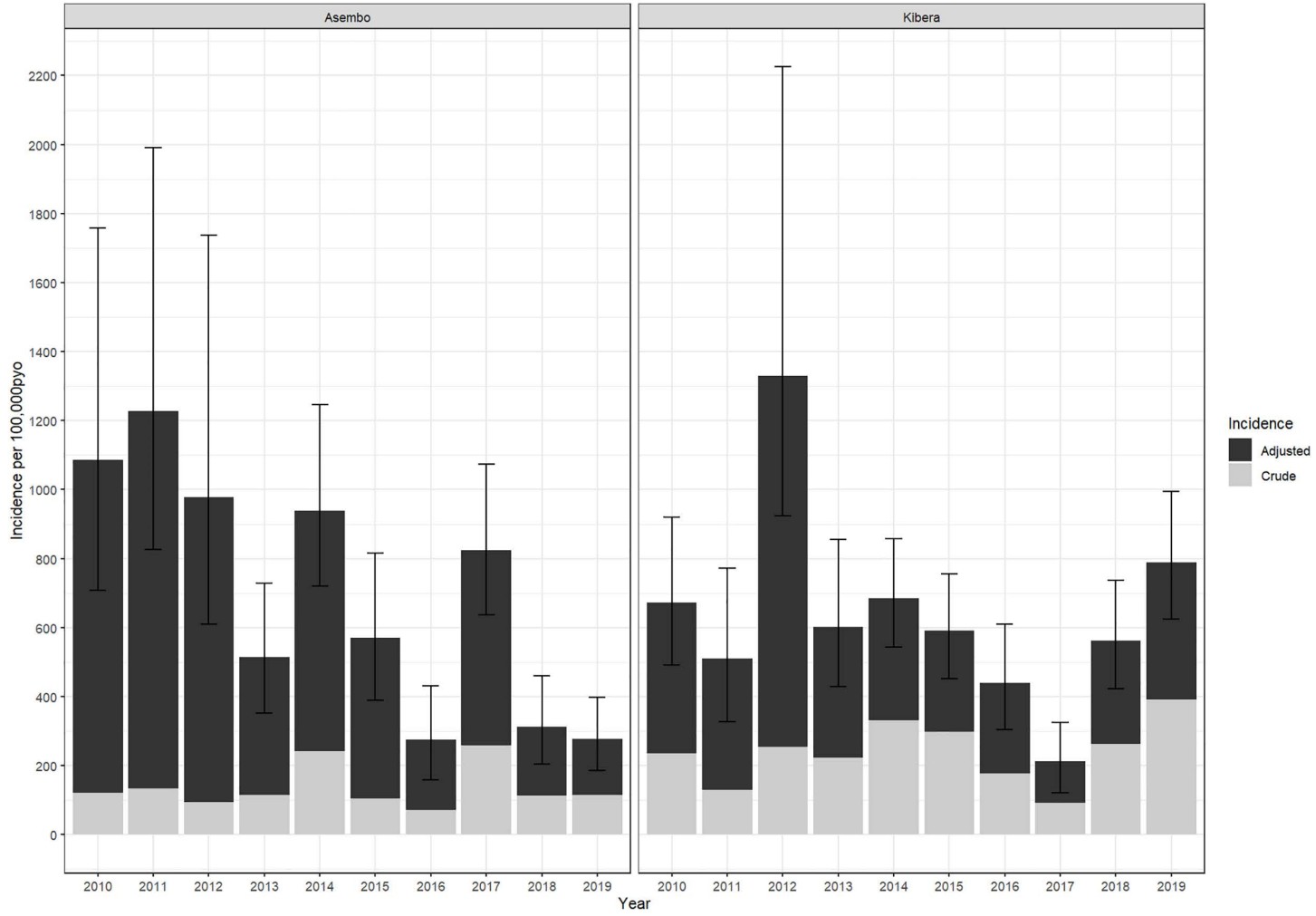

**Fig 4. Crude and Adjusted Annual *Shigella* Incidence in PBIDS: 2010-2018.** *Incidence rates were adjusted first for the proportion who were sampled among diarrhea cases, and second for proportion of cases with diarrhea at home who sought care in any clinic other than the surveillance clinic.

The high shigellosis burden observed among children aged 12–23 months is consistent with data from other resource-limited settings [11,17–19]. Several factors may contribute to a higher risk of shigellosis in the second year of life. Malnutrition, which is prevalent among children in this age group in resource limited settings [20], heightens the risk of diarrhea, both in terms of incidence and severity [21]. *Shigella*, specifically, is strongly associated with moderate-to-severe diarrhea in malnourished children [22]. Additionally, children aged 12–23 months have increasing interaction with possibly contaminated environmental reservoirs, especially in settings with inadequate WASH [20]. Moreover, less consumption of breastmilk in the second year of life means reduced protection from maternal antibodies and possibly lactoferrin, which may have antimicrobial activity against *Shigella* [23]. The high burden of disease among children aged 12–23 months suggests that a *Shigella* vaccine administered late in infancy [24,25] could be an important prevention tool, in conjunction with efforts to improve WASH.

We also observed a high burden of disease among elderly populations in the rural setting, but not among their urban counterparts. This finding may reflect higher HIV prevalence in the rural population [8,26,27], which has been linked to increased risk and severity of shigellosis [28,29].HIV infection among shigellosis cases in Asembo, based on available

**Fig 5. Antimicrobial drug resistance in *Shigella* among patients seeking care at PBIDS Sentinel Clinics, 2010-2018.**

HIV status data, was relatively higher than estimates of HIV in the general population from a national survey (25% vs 15%), suggesting possible association between HIV status and shigellosis. Ongoing efforts to prevent HIV infection and improve access and adherence to antiretroviral therapy in Kenya may therefore also serve to lessen the burden of shigellosis, particularly among older adults, and may help explain, in part, the decline in shigellosis incidence observed in the rural area only.

Antibiotics play an important role in the control of *Shigella* by reducing the duration of illness and risk of disease spread [30]. WHO recommends empiric treatment of dysentry with antibiotics, including ciprofloxacin and ceftriaxone as first line and second line drugs, respectively [31]. In our current study, non-susceptibility to ciprofloxacin and ceftriaxone was detected in both sites, albeit infrequently (<3% of isolates). The observed significant increase over time in resistance to nalidixic acid in Kibera is alarming because nalidixic acid resistance is often a precursor to ciprofloxacin resistance [32]. We observed poor adherence to WHO guidelines, including frequent antibiotic use for *Shigella* cases that presented with non-bloody stool, highlighting the unreliability of dysentery as a sole indicator of *Shigella*, and use of antimicrobials not

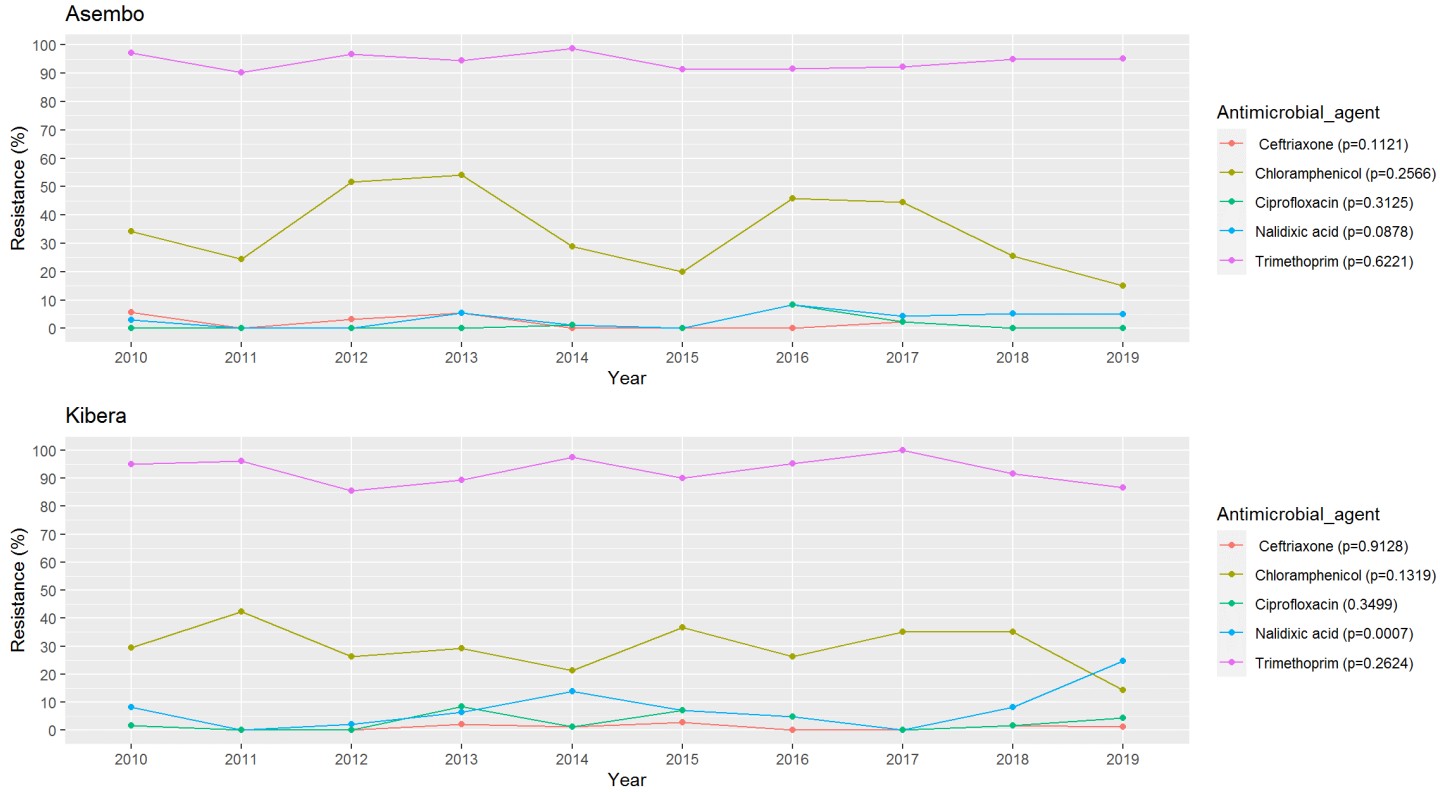

**Fig 6. Antimicrobial drug resistance trends in *Shigella* among patients seeking care at PBIDS Sentinel Clinics, 2010-2018.**

currently recommended for treatment of shigellosis (i.e., metronidazole, erythromycin). Furthermore, ~2% of patients were treated with drugs to which their *Shigella* isolates were not susceptible to. However, in the Vaccine Impact on Diarrhea in Africa (VIDA) study, as much as 25% of moderate-to-severe diarrhea cases in Kenya were prescribed an antibiotic when there was no indication [33]. These findings draw attention to the need for educating clinicians on guidelines for diarrhea management and of the need for continued monitoring of *Shigella* resistance patterns to help inform treatment recommendations.

Increasing evidence— including from this study, where 53% and 64% of *Shigella* cases in Asembo and Kibera, respectively, presented without bloody stool—suggests that dysentery alone, as stipulated in the existing WHO guidelines, may not be a reliable indicator of *Shigella* infection [34,35]. From these observational studies, watery shigellosis are potentially being missed and inappropriately treated, highlighting the need to reassess WHO guidelines and promote point-of-care Shigella diagnostics for accurate and cost-effective treatment [36].

*Shigella* vaccines under development offer promise in reducing disease burden and antibiotic resistance by preventing infection and limiting empiric antibiotic use. In both rural and urban Kenya, *S. flexneri*—responsible for about two-thirds of cases—was the most common species, aligning with global findings from low- and middle-income countries [37]. A vaccine targeting *Shigella flexneri* 2a, 3a, 6, and *Shigella sonnei* O-antigens could directly cover 64% of global *Shigella* isolates, based on serotyping data from the Global Enterics Multicenter Study, potentially reaching 88% coverage with cross-protection against other S. *flexneri* serotypes [38]. Our data highlight the need for a vaccine that offers protection through the second year of life aligning with WHO's Preferred Product Characteristics for *Shigella* vaccines targeting

infants and young children in LMICs [25]. Our data also suggest that in certain settings, vaccination of older adults against *Shigella* could be important for optimal disease control as they may be potential reservoirs of infection for children, especially in rural settings where children sometimes remain under the care of the elderly. *Shigella*-powered studies like Enterics for Global Health (EFGH) surveillance are crucial for generating up-to-date *Shigella*-specific incidence data and defining primary endpoints, thereby accelerating vaccine trials and licensure [39].

This study has limitations. Adjusted incidence estimates assume that untested diarrhea cases—both at non-surveillance facilities and among those without stool collection at the surveillance clinic—had similar *Shigella* detection rates as those tested, which may be incorrect. The low stool collection rate and differences between those who provided stool and those who did not may introduce selection bias. Changes in the frequency of household data collection over time could affect the accuracy of healthcare-seeking adjustments. Additionally, culture-based detection likely underestimates *Shigella* burden due to its lower sensitivity compared to molecular methods [18,20]. Furthermore, some subgroup estimates, particularly in Kibera, have wide confidence intervals and should be interpreted cautiously. No antibiotic susceptibility testing was conducted for azithromycin, which is also currently recommended for shigellosis. Furthermore, data on HIV status was missing for a substantial proportion of cases.

In conclusion, our findings demonstrate a high burden of *Shigella* among children aged 12–23 months in Kenya, as well as a relatively high burden among older adults and a declining burden over time in a rural context. *Shigella* isolates remain largely susceptible to first and second-line recommended antibiotics. However, continued monitoring is necessary to detect and contain possible emergence of resistant strains. Our findings suggest that *Shigella* vaccines under development, if successful, could help reduce both the incidence and antimicrobial resistance burden of *Shigella* in Kenya, and that longitudinal population-based surveillance data can inform the development of candidate vaccines, strategies for vaccine roll-out, and evaluation. Nonetheless, population-based surveillance should integrate data on household WASH infrastructure and practices to better identify risk factors and strengthen intervention strategies for long-term, definitive control of *Shigella* and other enteric infections.

## Disclosure

The findings and conclusions in this report are those of the authors and do not necessarily represent the official position of the Kenya Medical Research Institute, US Centers for Disease Control and Prevention, nor any other partner institutions.

## Supporting information

**S1 Fig. Population Distribution of PBIDS surveillance sites and sentinel clinics, Kenya 2010–2019.**
(TIF)

**S1 Table. Characteristics of diarrhoea cases that had stool sample collected compared to those who had no stool sample collected from PBIDS, 2010–2019.**
(DOCX)

**S2 Table. Distribution of *Shigella* species over time: 2010–2019.**
(DOCX)

**S3 Table. Age-stratified *Shigella* incidence in Asembo and Kibera surveillance areas, 2010–2019.**
(DOCX)

**S4 Table. *Shigella* incidence rates in Asembo and Kibera, 2010–2019.**
(DOCX)

## Acknowledgments

This study was jointly supported by the US Centers for Disease Control and Prevention (CDC) and the Kenya Medical Research Institute. We wish to thank the study participants in both Lwak Mission Hospital, Asembo and Tabitha Medical Clinic, Kibera, PBIDS staff and CDC staff based in Atlanta and Kenya for supporting this study. This paper has been approved for publication by the Director of the Kenya Medical Research Institute. The data used in this study are owned by KEMRI and are available from the corresponding author upon reasonable request, subject to institutional data sharing policies.

## Author contributions

**Conceptualization:** Richard Omore.

**Data curation:** Billy Ogwel.

**Formal analysis:** Billy Ogwel, Allan Audi, George O. Agogo.

**Funding acquisition:** Daniel R. Feikin, Marc-Alain Widdowson, Robert F. Breiman, Godfrey M. Bigogo, Jennifer R. Verani.

**Investigation:** Richard Omore, John B. Ochieng, Godfrey M. Bigogo, Jennifer R. Verani.

**Methodology:** Richard Omore, Billy Ogwel, John B. Ochieng, Jane Juma, Victor Omballa, Alice Ouma, George Aol, Allan Audi, George O. Agogo, George Odongo, Clayton Onyango, Newton Wamola, Terry Komo, Daisy Chepkemoi, Elizabeth Hunsperger, Daniel R. Feikin, Joel M. Montgomery, Marc-Alain Widdowson, Graeme Prentice-Mott, Eric D. Mintz, Robert F. Breiman, Patrick K. Munywoki, Godfrey M. Bigogo, Jennifer R. Verani.

**Supervision:** George Aol.

**Writing – original draft:** Richard Omore.

**Writing – review & editing:** Richard Omore, Billy Ogwel, John B. Ochieng, Jane Juma, Victor Omballa, Alice Ouma, George Aol, Allan Audi, George O. Agogo, George Odongo, Clayton Onyango, Newton Wamola, Terry Komo, Daisy Chepkemoi, Elizabeth Hunsperger, Daniel R. Feikin, Joel M. Montgomery, Marc-Alain Widdowson, Graeme Prentice-Mott, Eric D. Mintz, Robert F. Breiman, Patrick K. Munywoki, Godfrey M. Bigogo, Jennifer R. Verani.

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
