## [Editor Report · Decision Letter 0]

5 Nov 2025

Dear Dr. Omore,

Thank you for submitting your manuscript to PLOS ONE. After careful consideration, we feel that it has merit but does not fully meet PLOS ONE’s publication criteria as it currently stands. Therefore, we invite you to submit a revised version of the manuscript that addresses the points raised during the review process.

**ACADEMIC EDITOR: As attached document with track**

We look forward to receiving your revised manuscript.

Kind regards,

silas onyango awuor, msc

Academic Editor

PLOS ONE

“This study was funded by the US Centers for Disease Control and Prevention (CDC) in Atlanta USA through a cooperative agreement with KEMRI.”

“This study was funded by the US Centers for Disease Control and Prevention (CDC) in Atlanta USA through a cooperative agreement with KEMRI.”

“This study was funded by the US Centers for Disease Control and Prevention (CDC) in Atlanta USA through a cooperative agreement with KEMRI.”

4. In the online submission form you indicate that your data is not available for proprietary reasons and have provided a contact point for accessing this data. Please note that your current contact point is a co-author on this manuscript. According to our Data Policy, the contact point must not be an author on the manuscript and must be an institutional contact, ideally not an individual. Please revise your data statement to a non-author institutional point of contact, such as a data access or ethics committee, and send this to us via return email. Please also include contact information for the third party organization, and please include the full citation of where the data can be found.

Additional Editor Comments:

Kindly see the attached document with track for your action

---

## [Author Response · Author response to Decision Letter 1]

12 Nov 2025

November 09, 2025

To: Prof. Emily Chenette

Editor-in-Chief, PLOS One

1875 Mission Street

Suite 103 #188 San Francisco, CA 94103

United States

Dear Prof. Emily,

Please find attached our revised manuscript (#PONE-D-25-40297) entitled, “Population-based incidence and antimicrobial susceptibility patterns of shigellosis among children and adults from rural and urban Kenya, 2010-2019” submitted for consideration as an original article in PLOS One.

We are grateful to you and the reviewers for the thoughtful and constructive feedback. We have revised the manuscript in response to all comments, which we also wish to thank you since it has has improved its clarity and rigor.

A detailed, point-by-point response to each comment is included below, referencing the corresponding changes and line numbers in the revised, tracked version of the manuscript.

Thank you again for considering our resubmission. We appreciate the opportunity to improve our work and look forward to your positive evaluation.

Yours sincerely,

On behalf of the co-authors,

*Correspondence: Richard Omore; KEMRI-CGHR, P.O Box 1578-40100, Kisumu, Kenya

E-mail: omorerichard@gmail.com; Phone: +254-728-813-788

Point-by-point response to Editorial and Reviewer Comments

We sincerely appreciate the invaluable comments and suggestions provided by the editors, which we believe have contributed in improving the quality of this work. Below, we present a point-by-point response to each of the remarks, indicating the corresponding revisions made in the manuscript and referencing the relevant line numbers in the tracked version.

Editorial Comments to Author:

Comment# 1: Please ensure that your manuscript meets PLOS ONE's style requirements, including those for file naming. The PLOS ONE style templates can be found at

Response to comment #1: Thank you for this comment. We have revised the manuscript to align with the journal’s style requirements. Specifically, we reformatted Level 1 and Level 2 headings, bolded figure captions and table titles, confirmed file naming convention, and removed funding and competing interests information from the Acknowledgments section as suggested.

Comment #2: Thank you for stating the following financial disclosure:

“This study was funded by the US Centers for Disease Control and Prevention (CDC) in Atlanta USA through a cooperative agreement with KEMRI.”

Response to comment #2: Thank you for this comment. We have updated the financial disclosure statement, which now reads: “This study was funded by the US Centers for Disease Control and Prevention (CDC) in Atlanta USA through a cooperative agreement with KEMRI. The funders had no role in study design, data collection and analysis, decision to publish, or preparation of the manuscript.”. We have included the amended funder statement in the cover letter as suggested.

Comment #3: Thank you for stating the following in the Funding Section of your manuscript:

“This study was funded by the US Centers for Disease Control and Prevention (CDC) in Atlanta USA through a cooperative agreement with KEMRI.”

“This study was funded by the US Centers for Disease Control and Prevention (CDC) in Atlanta USA through a cooperative agreement with KEMRI.”

Response to comment #3: We appreciate this comment. We have now removed funding and competing interests information statements from the Acknowledgments section in the revised version as per your guidelines. We have included the amended funder statement in the cover letter.

Comment #4: In the online submission form you indicate that your data is not available for proprietary reasons and have provided a contact point for accessing this data. Please note that your current contact point is a co-author on this manuscript. According to our Data Policy, the contact point must not be an author on the manuscript and must be an institutional contact, ideally not an individual. Please revise your data statement to a non-author institutional point of contact, such as a data access or ethics committee, and send this to us via return email. Please also include contact information for the third-party organization, and please include the full citation of where the data can be found.

Response to comment #4: Thank you for this comment. We have revised the data availability statement accordingly (Lines 332-335) and now reads as follows:

“The data used in this study are owned by the Kenya Medical Research Institute (KEMRI). Access to the data may be granted upon reasonable request to the Head of the KEMRI Scientific and Ethics Review Unit (Email: seru@kemri.go.ke or serukemri@gmail.com), subject to KEMRI’s institutional data access and sharing policies.”

Comment #5: Please include captions for your Supporting Information files at the end of your manuscript, and update any in-text citations to match accordingly. Please see our Supporting Information guidelines for more information: http://journals.plos.org/plosone/s/supporting-information.

Response to comment #5: Thank you for this comment. We have now added the captions for the supporting information at the end of the revised manuscript (Lines 466-472).

Comment #6: If the reviewer comments include a recommendation to cite specific previously published works, please review and evaluate these publications to determine whether they are relevant and should be cited. There is no requirement to cite these works unless the editor has indicated otherwise.

Response to comment #6: We have not received any recommendations for citations at the moment, but we appreciate the guidance.

Additional Editor Comments:

Comment #7: To ensure your figures meet our technical requirements, please review our figure guidelines: https://journals.plos.org/plosone/s/figures

Response to comment #7: Thank you for this comment. We wish to confirm that the figures were generated using NAAS and meet the journal technical requirements.

---

## [Decision Letter · Decision Letter 1]

9 Mar 2026

Population-based incidence and antimicrobial susceptibility patterns of shigellosis among children and adults from rural and urban Kenya, 2010-2019

PONE-D-25-40297R1

Dear Dr. Omore,

We’re pleased to inform you that your manuscript has been judged scientifically suitable for publication and will be formally accepted for publication once it meets all outstanding technical requirements.

Kind regards,

Babak Pakbin

Academic Editor

PLOS One

Additional Editor Comments (optional):

Reviewers' comments:

Reviewer's Responses to Questions

**Comments to the Author**

Reviewer #1: All comments have been addressed

Reviewer #2: All comments have been addressed

2. Is the manuscript technically sound, and do the data support the conclusions?

Reviewer #1: Yes

Reviewer #2: Yes

3. Has the statistical analysis been performed appropriately and rigorously?

Reviewer #1: Yes

Reviewer #2: Yes

4. Have the authors made all data underlying the findings in their manuscript fully available?

Reviewer #1: Yes

Reviewer #2: No

5. Is the manuscript presented in an intelligible fashion and written in standard English?

Reviewer #1: Yes

Reviewer #2: Yes

Reviewer #1: The manuscript presented to me deals with an important topic. The epidemiology of shigellosis has gradually begun to fall out of focus among European researchers. However, the disease continues to be active in industrially developed regions. The description of the problem and the guidelines for solving gaps are valuable; the manuscript is structured adequately to the topic, provides answers to the questions discussed. Correct statistical methods have been used. I highly appreciate the manuscript and, after the changes made, I propose that it be accepted for publication.

Reviewer #2: The manuscript is technically sound, the statistical methodology is rigorous, and the conclusions are well-supported by the presented data. I recommend this manuscript for publication following some minor revisions primarily concerning data availability protocols and grammatical corrections.

-The methodology is robust and appropriate for the study's epidemiological scope.

Disease Burden: The conclusion that children aged 12-23 months bear the highest burden is clearly supported by the adjusted incidence rates of 1,873/100,000 in Asembo and 2,828/100,000 in Kibera.

Species Distribution: The authors appropriately draw vaccine-related conclusions based on their finding that S. flexneri accounts for approximately two-thirds of the isolates in both settings (61% and 67%).

-Statistical Rigor:

The authors have demonstrated excellent epidemiological rigor by moving beyond crude incidence estimates.

The application of a two-step adjustment to account for the proportion of eligible participants who had stool collected and the proportion who sought out-of-network care is a major strength of this paper.

Calculating the uncertainty around the adjusted incidence using Monte Carlo simulations (10,000 simulations sampling from Poisson and binomial distributions) is a highly sophisticated and appropriate technique to establish 95% Confidence Intervals.

The use of the Mann-Kendall test to assess temporal trends is the correct statistical choice for this data.

-Data Availability:

The current Data Availability Statement notes that KEMRI owns the data, which is available upon reasonable request to the corresponding author. To fully comply with the PLOS Data policy, the authors should not rely solely on a single author as the point of contact.

Recommendation: Please provide a persistent, non-author institutional contact (such as an email address for KEMRI's data access or ethics committee) to ensure long-term data accessibility for future researchers.

**Do you want your identity to be public for this peer review?** For information about this choice, including consent withdrawal, please see our Privacy Policy

Reviewer #1: **Yes:** Valeri R Velev

Reviewer #2: **Yes:** Ismail Yosri Abdelgelel Ismail

---

## [Editor Report · Acceptance letter]

PONE-D-25-40297R1

PLOS One

Dear Dr. Omore,

I'm pleased to inform you that your manuscript has been deemed suitable for publication in PLOS One. Congratulations! Your manuscript is now being handed over to our production team.

Kind regards,

on behalf of

Dr. Babak Pakbin

Academic Editor

PLOS One